# Conditional cash transfer and mortality among interpersonal violence victims: A cohort study

Camila Bonfim[1]*, Flávia Alves[1,2], Maurício L. Barreto[1], Vikram Patel[2], Daiane Borges Machado[1,2]

1 Centre of Data and Knowledge Integration for Health (CIDACS), Salvador, Bahia, Brazil, 2 Department of Global Health and Social Medicine, Harvard Medical School, Boston, Massachusetts, United States of America

* camila.bonfim@fiocruz.br

## Abstract

### Background

Interpersonal violence is a significant public health issue, increasing mortality risks for those affected. While Cash Transfer Programs offer health benefits, their role in addressing the needs of interpersonal violence victims remains unclear. This study aims to examine the association between Brazil's Bolsa Família Program (BFP) participation and reduced mortality rates among interpersonal violence victims.

### Methods and findings

This cohort study was conducted using data from 100 Million Brazilian Cohort, which were linked with interpersonal violence registries (2011−2015). All individuals with a record of interpersonal violence following their registration in Brazil's primary social assistance system during the study period were included. The primary outcome was overall mortality, while secondary outcomes comprised deaths due to natural and unnatural causes, as recorded in the Mortality Information System (SIM) and classified according to the International Classification of Diseases, 10th Revision (ICD-10). We used Cox proportional hazards models with propensity score-based method to analyze overall mortality and competing risk models to assess specific causes of death, estimating the association between BFP participation and mortality rates. A total of 29,075 individuals who were victims of interpersonal violence were followed throughout the five-year period. A total of 990 individuals died from overall causes. BFP participation was associated with an 18% reduction in overall mortality rate (hazard ratio, HR 0.82, 95% CI [0.70,0.95]; $p = 0.011$) and a 66% reduction in mortality rate from natural causes (HR 0.34 [95% CI 0.28, 0.41]; $p < 0.001$). This sample includes only individuals who seek healthcare services, which may overrepresent more severe cases of interpersonal violence.

**Data availability statement:** The information examined in this research is stored and managed by the Center for Data and Knowledge Integration for Health (CIDACS). Complete access to this data is limited due to its confidential nature and the specific licensing agreement that restricts its use to this study. Brazilian ethical guidelines, as determined by the Ethics Committee, prevent this data from being publicly shared. However, upon request if all ethical and legal criteria are satisfied, the data curation team at the institution may grant access to the data. For additional details, inquiries can be sent to cidacs.curadoria@fiocruz.br. The code used in the analysis is available from Github [https://github.com/profacamilabonfim/Do-file-of-the-paper-Conditional-cash-transfer-and-mortality-among-interpersonal-violence-victims.git] and archived in Zenodo [https://doi.org/10.5281/zenodo.18852494]".

**Funding:** The research presented in this manuscript was funded by the National Institute of Mental Health, part of the National Institutes of Health (URL: https://grants.nih.gov/), through Award Number R01MH128911 awarded to DBM. The funders had no role in study design, data collection and analysis, decision to publish, or preparation of the manuscript.

**Competing interests:** The authors have declared that no competing interests exist.

**Abbreviations:** AIC, Akaike Information Criteria; BFP, Bolsa Família Program; BIC, Bayesian Information Criteria; CIDACS, Center for Data Integration and Knowledge in Health; CTPs, Cash Transfer Programs; HR, hazard ratio; ICD-10, International Classification of Diseases, 10th Revision; IPTW, Inverse Probability of Treatment Weighting; LMICs, Low- and Middle-Income Countries; PS, propensity score; SIH, Hospitalization Information System; SIM, Mortality Information System; SINAN, National Disease Notification System; STROBE, Strengthening the Reporting of Observational Studies in Epidemiology.

## Conclusions

The association between BFP participation and lower mortality rates, especially from natural causes, among interpersonal violence victims highlights that such programs may be associated with reductions in poverty, improvements in health outcomes, and increased survival in vulnerable populations.

---

## Author summary
### Why was this study done?

- Interpersonal violence is a major public health problem associated with increased risk of morbidity and mortality.

- Although cash transfer programs are known to improve health outcomes and reduce social inequalities, their role in addressing the specific needs of victims of interpersonal violence remains unclear.

### What did the researchers do and find?

- This cohort study followed nearly 30,000 victims of interpersonal violence from the 100 Million Brazilian Cohort over a 5-year period.

- There was an association between the Bolsa Família Program (BFP) and a reduction in all-cause mortality (18%)

- This was primarily driven by a significant decrease in deaths from natural causes (66%).

### What do these findings mean?

- Our findings underscore the potential of cash transfers in reducing mortality rate among people victimized by interpersonal violence even when these programs were not specifically designed for them.

- Cash transfer programs could be used as a tool for prevention, reducing the mortality burden among victims of interpersonal violence.

- Hence, it is imperative for health administrators to advocate for the allocation of funds to support victims of interpersonal violence while concurrently addressing their health needs.

- The main limitation of this study is that it was restricted to individuals who sought healthcare services, which likely represent the most severe cases of interpersonal violence and may have led to misclassification bias.

## Introduction

Interpersonal violence is a multifactorial public health problem encompassing a wide range of relational contexts and manifestations. It may involve family members, intimate partners, friends, acquaintances, or strangers, and includes various forms such as child maltreatment, youth violence, violence against women, and elder abuse [1]. Its consequences extend across multiple domains, including physical, psychological, financial, sexual, neglect-related, and other dimensions [2,3].

It affects millions of people worldwide, ranking as the third leading cause of disability-adjusted life years and the second leading cause of years of life lost due to premature mortality in 2019 [4]. In addition to experiencing violence, these individuals also face reduced life expectancy [4]. However, what is associated with lower mortality among individuals exposed to violence is still not well-established [5].

This is a global issue, with a higher impact in countries such as Brazil, where the mortality rate due to violence is among the highest worldwide, reaching 22.4 per 100,000 inhabitants [5]. Although previous studies have estimated the global prevalence of interpersonal violence [4,6,7], assessing its health burden remains challenging due to its complexity and frequent underreporting, as many victims do not seek healthcare after violent episodes [2].

Interpersonal violence has been associated with social inequalities related to poverty, such as limited employment opportunities, restricted access to education, and gender, racial, and income disparities [2,5], highlighting heterogeneity according to the vulnerability of the at-risk population. These inequalities further increase vulnerability to the negative health outcomes observed in victims of interpersonal violence [8]. They are at higher risk of non-communicable diseases such as cardiovascular disease, cancer, respiratory problems, and diabetes as well as psychiatric disorders [2,9]. Exposure to violence can also lead to damage to the nervous, endocrine, and immune systems in addition to genetic changes associated with the environment [9]. Furthermore, the victims commonly have worse health habits such as alcohol abuse, tobacco use, and physical inactivity [2]. These factors contribute to increased mortality risk [4,6,10].

Considering interpersonal violence an important risk factor for poor health across the life course, preventing this public health problem has been a goal of various initiatives [2,6]. Programs that reduce social inequalities such as Cash Transfer Programs (CTPs) have had an association with decreased violence risk [2,5]. CTPs are social protection programs designed to reduce poverty and vulnerability through cash-based benefits [11]. They can be unconditional or conditional, the latter can include attendance at health appointments, access to social services, school attendance when the family have children or adolescents, prenatal appointments for pregnant and mandatory vaccinations for children [11].

CTPs such as Bolsa Família Program (BFP) have been associated with improved health outcomes among beneficiaries [12]. This is one of the largest conditional cash transfer initiatives globally [12]. Since its implementation in 2004, it has played a significant role in reducing poverty and extreme poverty among Brazilian families [13]. Although originally designed as a social equity policy, evidence indicates that the BFP is also associated with improvements in health outcomes and lower mortality [12]. These associations may reflect not only the income transfer itself but also from the program's conditionalities [12]. By providing financial support to families, the program contributes to improved living conditions and fosters engagement with healthcare services and school attendance [12].

It is already known that these programs, especially conditional CTPs, may be associated with lower mortality among people hospitalized with psychiatric disorders [14], homicide [15], and suicide [16] rates in the general population. However, whether conditional CTPs would reduce mortality rates among people victimized by interpersonal violence, remains unanswered.

The role of BFP in health and mortality has been studied, and its main mechanisms have been explored [12,17,18]. Regarding specific populations, such as individuals exposed to interpersonal violence, we believe that the role of BFP operates through secondary prevention, by facilitating referral to health services, as well as through the well-documented improvement of living conditions reported in other studies [12,19].

We hypothesized that the reduction in mortality may occur by improving access to health services and mitigating physiological stress responses to violence as well as facilitating connection to other social protection programs, promoting social inclusion, better living conditions, and improved physical and mental health [2,5].

These individuals face dual vulnerability, as they live in poverty [20] and have been victims of violence [4,5] both strong risk factors for mortality. The literature has significant limitations, as none of the available studies have evaluated competing risks of death in this specific population. Assessing this is crucial, as mortality risks vary depending on the context [21]. Therefore, this study aimed to test the association between receiving benefits from the Brazilian conditional CTP, the Bolsa Familia Program (BFP), and decreased mortality rates among victims of interpersonal violence. We hypothesized that BFP would be associated with a reduction in mortality rates, controlling for competing death risks, among these individuals.

## Method

### Study design, setting, and data source

This study followed a prospective protocol, with detailed descriptions of the study design and procedures published elsewhere [22]. This study used a cohort design from the 100 million Brazilian Cohort baseline, a dynamic cohort developed by the Center for Data Integration and Knowledge in Health (CIDACS) [23,24] based on a non-deterministic linkage between administrative health and social assistance databases [25]. The 100 million Brazilian Cohort baseline was developed to investigate social determinants and the impact of social programs and policies on various health contexts in Brazil [24]. This dataset is based on information from over 131 million individuals who were registered between 2001 and 2,018 in CadÚnico, the primary system for applying for social benefits in Brazil which include the BFP [23]. It includes socio-economic and sociodemographic information from poorer Brazilian individuals and their families who apply for social programs such as BFP [24]. To qualify and register with CadÚnico, families must have a per capita income of up to half a minimum wage or a total family income of up to three minimum wages [24].

We selected a subset of just over 23 million for this study based on the period during which all data on interpersonal violence and BFP were available (January, 1, 2011, to December, 31, 2015). The methods and analyses were detailed in the research protocol published previously [22]. This study is reported as the Strengthening the Reporting of Observational Studies in Epidemiology (STROBE) guideline (S1 STROBE Checklist).

Interpersonal violence records were extracted from two administrative databases, the National Disease Notification System (SINAN) (Method A in S1 Appendix) and the Hospitalization Information System (SIH) (Method A in S1 Appendix). SINAN is the system for mandatory reporting of interpersonal violence since 2011 [26,27]. This system includes physical, sexual, psychological, psychological abuses, neglect and other relational contexts perpetrated by family members, intimate partners, friends, or strangers [26]. The data is registered by professionals working in health facilities, and it has improved its reporting over the years.

Individuals hospitalized due to interpersonal violence were also identified through the SIH. SIH is the system responsible for recording most of Brazilian hospital admissions [28,29], including those due to violent causes, according to the International Classification of Diseases, 10th Revision (ICD-10) [30]. In this study, we included hospital admissions for aggression (codes X85-Y09) which are recorded as secondary diagnoses, as well as other external causes. However, to capture all possible hospitalizations related to this cause within the system, we also include records where it appears as the primary diagnosis.

We also used Mortality Information System (SIM), which comprises all deaths in the country and uses a mandatory certificate internationally recognized as a high-quality system [31] (Method A in S1 Appendix). SIM registers the cause of death according to the ICD-10 [32]. All the health systems use standardized forms completed by health professionals [22].

Finally, this study extracted information from the BFP database, the main poverty and extreme poverty alleviation program implemented by the Brazilian government in 2004 [24]. All participants benefiting from the BFP are registered in the

CadÚnico. These databases were linked using a tool developed by CIDACS, which uses five identifiers (name, gender, year of birth, name of the mother and municipality of residency) [22]. Reliability analyses showed very high sensitivity and specificity of the linkage [25] (SMethod B in S1 Appendix). Additional information on data governance and the linkage process has been published elsewhere [25]. This study was approved by the ethics committees of the Federal University of Bahia (UFBA – registration number: 1023107) and Gonçalo Muniz Institute at the Oswaldo Cruz Foundation (registration number: 1.612.302).

## Participants

Individuals from the 100 million Brazilian cohort who were reported in SINAN ($N = 67,487$) or SIH ($N = 11,286$) as having experienced interpersonal violence ($N = 78,038$) from January 1, 2011, to December 31, 2015. The SINAN system collects all violence-related information reported in the country.

To ensure an unbiased sample, we focused on those who became victims of violence without prior exposure to program intervention. Therefore, we included only individuals who were registered in CadÚnico after a violent event ($N = 30,323$). Individuals who were already receiving the benefit before the violent event may have a lower chance of hospitalization or notification of violence, considering the association between receiving the benefit and improved health [14–18,32,33]. As individuals with multiple records of interpersonal violence may have a higher risk of mortality [2], we removed duplicate entries (1,174 cases, 4%) to mitigate biases in the association measurement. Finally, we removed individuals who had inconsistent data on dates, such as death before CadÚnico registration or the same start and end date for receiving BFP ($n = 74$, < 1% of the total sample). Therefore, the study included 29,075 participants, of whom 14,856 (51%) were BFP recipients (Fig 1).

## Follow-up

For the subset of BFP beneficiaries, (A) individuals were followed from the moment they received the benefit after notification/hospitalization, starting on January 1, 2011. Follow-up ended at the earliest occurrence of either: (B) the individual's death from any cause, or (C) December 31, 2015. For the subset of non-beneficiaries, (A) follow-up began from the moment they were registered in CadÚnico following notification/hospitalization, starting on January 1, 2011. Follow-up ended at the earliest occurrence of either: (B) the individual's death from any cause, or (C) December 31, 2015.

## Variables

**Exposure.** The Brazilian cash transfer Program BFP aims to lift people out of extreme poverty or poverty [34]. In addition to providing income to families in poverty, the BFP aims to integrate public policies, improve access to basic rights such as health and education [34]. The benefits range from BRL 41.00 (USD 10.00) to BRL 300.00 (USD 75.00) per individual, based on 2015 values adjusted for inflation [22]. Eligibility depends on a monthly household income of less than BRL 70.00 (USD 17.00), or BRL 140.00 (USD 34.00) for households with a child, teenager, or pregnant woman [23]. Recipients of the BFP must meet specific conditionalities to maintain their benefits, including minimum school attendance, keeping up with vaccinations, and monitoring the growth of young children [23]. Additionally, pregnant or breastfeeding women are required to follow a specified health and nutrition protocol [23]. The conditionalities are grounded in the notion that tying benefits to constructive behaviors can further improve families' chances of breaking free from poverty through educational attainment or health improvements [23].

We considered the beneficiary group (exposed) individuals who receive the benefit after the interpersonal violence event over the period of the study. The non-beneficiary group (unexposed) was composed of individuals who were also victims of interpersonal violence and did not receive the benefit during the same period. We highlighted that both groups were identified through the CadÚnico. This system includes individuals who meet the criteria for poverty or extreme poverty and who may apply for various social benefits such as social benefits for people with disabilities, housing for

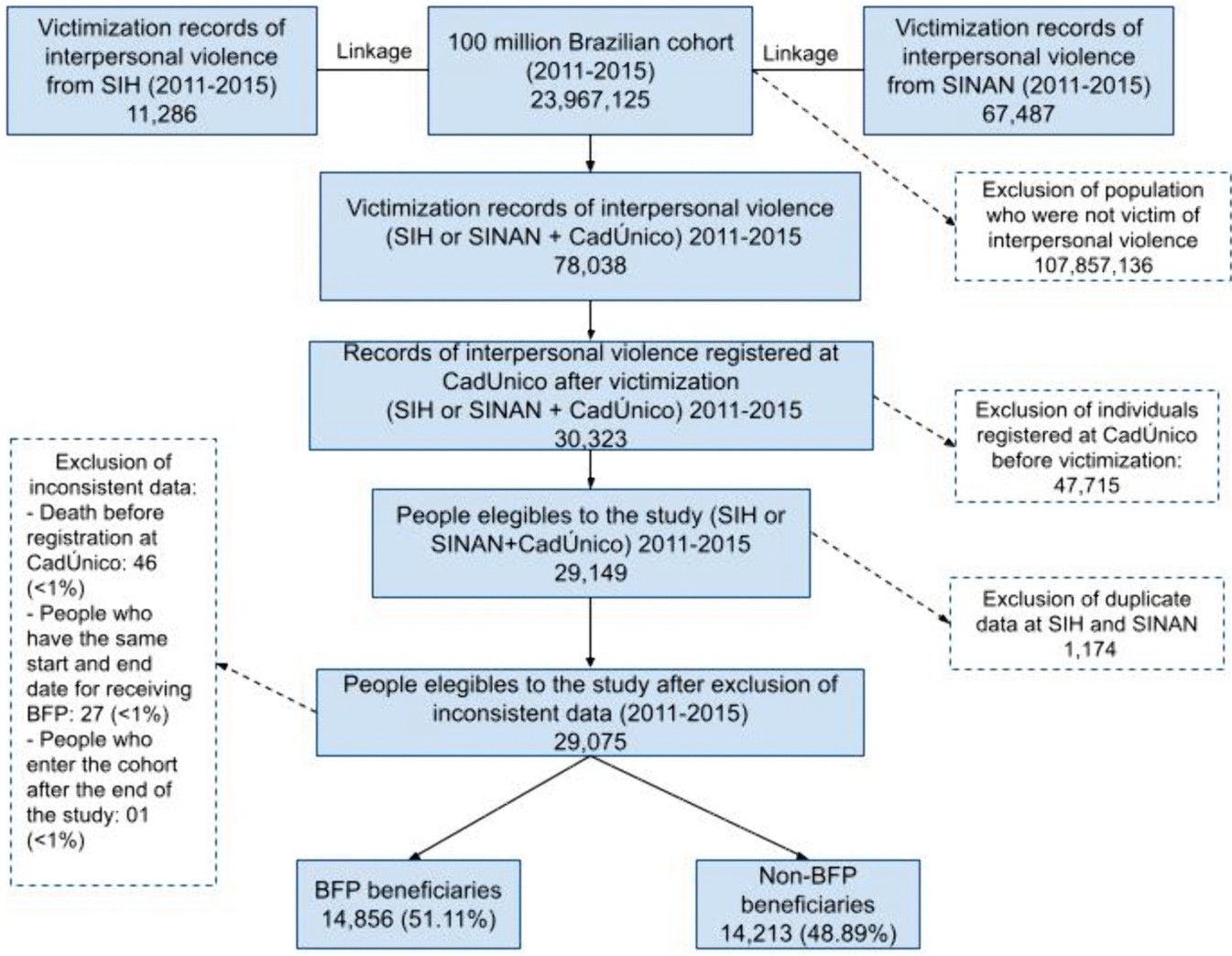

**Fig 1. Flowchart of the study population.** Abbreviations: SIH, Hospitalization Information System (acronymous in Portuguese); SINAN, National Disease Notification System (acronymous in Portuguese); BFP, Bolsa Familia Program.

low-income families, BFP, among others [19]. BFP has more stringent eligibility standards and focuses on families that constitute a defined subset within CadÚnico [23]. Therefore, not all individuals meet the eligibility criteria for participation in the BFP.

**Outcomes.** Our primary outcome was a record of overall mortality in the SIM. Secondary outcomes included natural and unnatural causes of death. Natural causes of death were defined as all causes of mortality, except for suicide and external causes (X60-Y09), according to ICD-10 [30]. Unnatural causes included external causes such as accidents, homicide, suicide, and other external causes of death.

## Statistical analysis

Following our study protocol [22] and other studies using the 100 million Brazilian cohort [14–18,32,33], we used a propensity score (PS)-based method [35,36] to identify the association between BFP and reduction of mortality rates.

By applying stabilized IPTW, we aimed to balance differences between groups [35]. Although this cohort contains complete data, beneficiaries and non-beneficiaries may still differ because eligibility for the BFP depends on income and additional socioeconomic criteria that may introduce imbalance [23]. The use of PS estimation allowed us to model the probability of receiving the BFP based on observable baseline sociodemographic characteristics, thereby reducing potential selection bias, especially given that eligibility could not be fully determined using income cutoff alone. This methodological approach is consistent with prior studies [14–18,32,33].

First, we ran a logistic regression to estimate the PS using baseline covariates, which are associated with BFP according to the literature review [14–18,32,33] (Table A in S1 Appendix). The following covariates were considered to estimate the PS: sex, age, education level, race, location of residence (rural × urban), living conditions (which included water supply, waste, sanitation, and construction materials), isolation (single person in the household), Brazilian regions of residence, and year of CadÚnico registration. We also evaluated the common support graph and compared the range of PS between the BFP and non-BFP groups (Fig A in S1 Appendix).

Second, the stabilized Inverse Probability of Treatment Weighting (IPTW) approach was applied [35]. We applied the stabilized IPTW weights for non-beneficiaries using the formula $(1 − Pt)/(1 − Psmul)$ and for beneficiaries using the formula $Pt/Psmul$, where "Pt" is the marginal probability of treatment in the population and "Psmul" is the PS obtained from the multivariable logistic regression adjusted for covariates [35]. To improve the accuracy and the precision of the estimation, in addition to stabilized IPTW, weight truncation was performed based on distribution for the 99th percentile [14,32,35].

Third, standardized differences among covariates of beneficiaries and non-beneficiaries before and after applying stabilized IPTW weighting were calculated to assess covariate balance, and changes in absolute values greater than 10% were considered acceptable [36].

Finally, we used stabilized IPTW-weighted Cox proportional hazards regression analysis, adjusted for the year of notification or hospitalization due to interpersonal violence, to examine the association between BFP and the overall mortality rate in the final model [35,36]. To address the potential bias of overestimating the risk of the event of interest due to competing risks in traditional Cox regression, we employed competing risk models using the Fine-Gray subdistribution hazard model [37]. For this analysis, each specific outcome was considered a failure, other causes were treated as competing risks, and individuals who remained alive were censored. A stratified analysis was also performed by sex, considering that gender differences [2,5] may influence mortality risk (TableB in S1 Appendix).

## Sensitivity analyses

We conducted the following analysis to assess the robustness of the results. First, to test whether our results are not an artifact created with the stabilized IPTW weights, we used another PS-based method, the Kernel Matching approach [35] (Table 3). Second, we repeated the analysis using Poisson models with Incidence Rate Ratios estimation and 95% CI (Table C in S1 Appendix). Third, to address potential selection bias, we conducted the analysis with the entire population victimized by interpersonal violence during the study period, including those registered in CadÚnico before the violent event (Table D in S1 Appendix). The main analyses were additionally replicated by treating missing data as a separate category, with the aim of assessing potential biases arising from information loss (Table E in S1 Appendix). Furthermore, we ran crude Cox model without propensity adjustment to evaluate if the results are an artifact created with the IPTW weights (Table F in S1 Appendix).

Fourth, we ran Cox proportional hazards models with time-varying covariates [38] considering BFP as a time-varying covariate to account for possible differences in the duration of benefit receipt and immortal time bias [38] (Table G in S1 Appendix). Fifth, to assess whether age modified the association between exposure and mortality [39], we compared models with and without an age–exposure interaction term using the Akaike (AIC) and Bayesian (BIC) Information Criteria. According to established guidelines, an AIC difference of less than two indicates no substantial improvement in model fit

[40] (Table H in S1 Appendix). Finally, to check the broad age range within the 25–59-year category of age, we conducted an additional sensitivity analysis introducing a more granular age stratification (Table I in S1 Appendix). Stata version 15.0 was used for the data analysis.

## Results

We identified 78,038 records of interpersonal violence in either the SIH or SINAN databases between 2011 and 2015 in the baseline of the study. After applying the exclusion and inclusion criteria, the study sample comprised 29,075 individuals who were registered at CadÚnico following a single hospitalization or notification of interpersonal violence (Fig 1). Most of these records were in SINAN (78.75%). Of these, 14,856 (51.11%) were BFP beneficiaries.

The average time of receiving BFP after the violent event was 1.71 years (SD = 1.10). Before weighting using stabilized IPTW, beneficiaries, compared to non-beneficiaries, were more likely to be female, aged 25–59 years old, had a high school level of education, were brown/ mixed race, lived in urban areas and in Southeast Brazil, had good household conditions, lived with someone else and were registered at CadÚnico in 2014. After using the stabilized IPTW, the groups became more balanced in most of the covariates (Table 1).

During the follow-up, 990 deaths were identified, the majority of which were due to natural causes in both BFP groups (BFP: 33.71%; Non-BFP: 66.29%; $p < 0.001$). BFP beneficiaries had lower mortality rates for: overall mortality (1349.04 [95% CI 1223.55, 1487.39]) and natural causes (796.70 [95% CI 701.64, 904.63]), both estimated for 100,000 inhabitants. Females and younger beneficiaries showed lower mortality rates except for unnatural causes of death (Table 2).

The receipt of BFP benefits was associated with a lower overall mortality (HR, hazard ratio 0.82 [95% CI 0.70,0.95]; $p = 0.011$). Using the Fine-Gray subdistribution hazard model, the subdistribution hazard ratio for natural causes was 0.34 ([95% CI 0.28,0.41]; $p < 0.001$), while that for unnatural causes of death was 0.90 ([95% CI 0.67,1.19]; $p = 0.461$) among the beneficiary group. Therefore, the BFP was associated with a 66% decrease in the natural causes of deaths and a 10% decrease for unnatural causes (non-significant) among individuals who either experienced a competing risk or were still alive. Robustness checks using time-varying and stabilized IPTW Cox models confirmed these findings, suggesting a lower risk of mortality among BFP beneficiaries (Table D in S1 Appendix).

The stratified analysis by sex showed that the BFP reduced mortality mainly among women, including both all-cause (HR 0.42 [95% CI 0.34,0.52]; $p \leq 0.001$) and natural causes (HR 0.61 [95% CI 0.46,0.80]; $p = 0.001$), whereas among men, the reduction was observed only for natural causes (HR 0.69 [95% CI 0.52,0.91]; $p = 0.009$) (Table B in S1 Appendix).

In general, the sensitivity analyses confirmed the findings (Tables 3 and A–I in S1 Appendix).

## Discussion

This is the first study to estimate the role of a conditional CTP on mortality among individuals with a documented history of interpersonal violence. Using longitudinal data from over 29,000 low-income individuals followed for up to five years, we observed an 18% reduction in overall mortality and a 66% reduction in mortality from natural causes as well as 10% from unnatural causes (non-significant) among program participants. These findings support the hypothesis that CTPs, primarily designed to alleviate poverty, can also improve the living conditions of individuals who have experienced violence, thereby preventing health deterioration and reducing mortality.

While there is some divergence in the literature regarding the role of conditional CTPs on violence [5], the role of such programs on mortality in the general population is well-established [5,41,42]. A multinational study that used data from 7 million people living in 37 Latin American countries showed that CTPs reduced overall mortality [43].

While preventing violence remains a global challenge, identifying measures that mitigate adverse outcomes among victims is crucial. To our knowledge, no previous studies have specifically evaluated the contribution of CTPs on mortality in populations who have been victims of interpersonal violence. Our study addresses a critical evidence gap by providing

**Table 1. Study population characteristics overall and by Bolsa Família Program participation before and after applying stabilized IPTW, 2011–2015.**

| Characteristics | Participants, No. (%) | Before IPTW N = 29,075 | | | After IPTW N = 26,093 | | |
|---|---|---|---|---|---|---|---|
| | Overall N = 29,075 | BFP n = 14,859 (51.11%) | Non-BFP n = 14,216 (48.89%) | Diff[1] (BFP − non-BFP) | BFP n = 12,884 (49.38) | Non-BFP n = 13,209 (50.52) | Diff[1] (BFP − non-BFP) |
| | N (%) | N (%) | N (%) | | N (%) | N (%) | |
| **Sex** | | | | | | | |
| Male | 11,387 (39.16) | 36.53 | 41.92 | 0.14 | 0.31 | 0.38 | 0.13 |
| Female | 17,688 (60.84) | 63.47 | 58.08 | | 0.69 | 0.62 | |
| **Age groups (years)** | | | | | | | |
| <10 | 7,686 (26.44) | 32.75 | 19.84 | −0.44 | 0.42 | 0.27 | −0.48 |
| 10–24 | 7,303 (25.12) | 26.72 | 23.45 | | 0.27 | 0.25 | |
| 25–59 | 12,440 (42.79) | 39.26 | 46.48 | | 0.30 | 0.42 | |
| >60 | 1,646 (5.66) | 1.28 | 10.24 | | 0.01 | 0.06 | |
| **Education Level (years of education)** | | | | | | | |
| Never been study | 5,306 (18.25) | 21.19 | 15.18 | −0.21 | 0.26 | 0.17 | −0.26 |
| Preschool | 1,929 (6.63) | 7.46 | 5.78 | | 0.08 | 0.07 | |
| Primary school or less (≤5 years) | 7,564 (26.02) | 25.16 | 26.91 | | 0.24 | 0.26 | |
| Junior high school (6–10 years) | 6,189 (21.29) | 22.10 | 20.43 | | 0.21 | 0.21 | |
| High school (10–12 years) | 7,538 (25.93) | 23.11 | 28.87 | | 0.20 | 0.27 | |
| College/university (≥13 years) | 502 (1.73) | 0.81 | 2.69 | | 0.01 | 0.02 | |
| Missing data | 47 (0.16) | 0.18 | 0.14 | | NA | NA | |
| **Race** | | | | | | | |
| White | 11,946 (41.09) | 37.80 | 44.52 | 0.09 | 0.36 | 0.43 | 0.11 |
| Black | 1,916 (6.59) | 7.13 | 6.02 | | 0.07 | 0.06 | |
| Asian | 165 (0.57) | 0.67 | 0.46 | | 0.01 | 0.01 | |
| Brown/ mixed race | 13,620 (46.84) | 49.05 | 44.53 | | 0.50 | 0.46 | |
| Indigenous | 138 (0.47) | 0.68 | 0.26 | | 0.01 | 0.01 | |
| Missing data | 1,290 (4.44) | 4.66 | 4.20 | | 0.04 | 0.05 | |
| **Location of residence** | | | | | | | |
| Rural | 2,317 (7.97) | 8.26 | 7.67 | 0.05 | 0.07 | 0.06 | 0.02 |
| Urban | 26,234 (90.23) | 88.70 | 91.83 | | 0.93 | 0.94 | |
| Missing data | 524 (1.80) | 3.04 | 0.51 | | NA | NA | |
| **Brazilian regions** | | | | | | | |
| Southeast | 12,037 (41.40) | 42.91 | 39.82 | 0.16 | 0.45 | 0.42 | −0.50 |
| Northeast | 4,197 (14.44) | 15.20 | 13.64 | | 0.15 | 0.14 | |
| Midwest | 3,575 (12.30) | 10.96 | 13.70 | | 0.11 | 0.13 | |
| South | 6,631 (22.81) | 20.98 | 24.72 | | 0.19 | 0.23 | |
| North | 2,635 (9.06) | 9.96 | 8.12 | | 0.10 | 0.08 | |
| **Household characteristics** | | | | | | | |
| **Water supply** | | | | | | | |
| Public network (running water) | 23,228 (78.89) | 76.28 | 83.66 | 0.09 | 0.84 | 0.88 | 0.12 |
| Well, natural sources, or other | 3,718 (12.79) | 14.32 | 11.18 | | 0.16 | 0.12 | |
| Missing data | 2,129 (7.32) | 9.39 | 5.16 | | NA | NA | |

*(Continued)*

**Table 1.** (Continued)

| Characteristics | Participants, No. (%) Overall N = 29,075 | Before IPTW N = 29,075 | | | After IPTW N = 26,093 | | |
|---|---|---|---|---|---|---|---|
| | | BFP n = 14,859 (51.11%) | Non-BFP n = 14,216 (48.89%) | Diff[1] (BFP − non-BFP) | BFP n = 12,884 (49.38) | Non-BFP n = 13,209 (50.52) | Diff[1] (BFP − non-BFP) |
| | N (%) | N (%) | N (%) | | N (%) | N (%) | |
| **Waste** | | | | | | | |
| Public collection system | 25,311 (87.05) | 84.43 | 89.79 | 0.07 | 0.94 | 0.95 | 0.05 |
| Burned, buried, outdoor disposal, or other | 1,635 (5.62) | 6.17 | 5.05 | | 0.06 | 0.05 | |
| Missing data | 2,129 (7.32) | 9.39 | 5.16 | | NA | NA | |
| **Sanitation** | | | | | | | |
| Public network | 16,958 (58.33) | 55.93 | 60.83 | 0.05 | 0.63 | 0.65 | 0.08 |
| Septic tank | 3,661 (12.59) | 11.87 | 13.34 | | 0.13 | 0.14 | |
| Homemade septic tank | 4,686 (16.12) | 15.60 | 16.66 | | 0.18 | 0.18 | |
| Ditch or other | 825 (2.84) | 3.43 | 2.22 | | 0.06 | 0.03 | |
| Missing data | 2,945 (10.13) | 13.16 | 6.96 | | NA | NA | |
| **Construction materials** | | | | | | | |
| Bricks/cement | 23,271 (80.04) | 76.55 | 83.68 | 0.08 | 0.94 | 0.95 | 0.10 |
| Wood, other plant materials, or other | 3,675 (12.64) | 14.05 | 11.16 | | 0.06 | 0.05 | |
| Missing data | 2,129 (7.32) | 9.39 | 5.16 | | NA | NA | |
| **Isolation** | | | | | | | |
| Live with someone else | 25,254 (86.86) | 89.37 | 84.23 | −0.24 | 0.95 | 0.89 | −0.23 |
| Live alone | 3,821 (13.14) | 10.63 | 15.77 | | 0.05 | 0.11 | |
| **Year of registration at CadÚnico** | | | | | | | |
| 2011 | 997 (3.43) | 4.58 | 2.22 | −0.42 | 0.07 | 0.03 | −0.46 |
| 2012 | 4,284 (14.73) | 18.54 | 10.76 | | 0.24 | 0.14 | |
| 2013 | 6,316 (21.72) | 25.22 | 18.06 | | 0.28 | 0.22 | |
| 2014 | 8,918 (30.67) | 29.62 | 31.77 | | 0.26 | 0.31 | |
| 2015 | 8,560 (29.44) | 22.03 | 37.18 | | 0.15 | 0.30 | |

[1]The difference in proportions of each category between BFP beneficiaries and non-beneficiaries (BFP beneficiary proportion minus non-beneficiary proportion).

Abbreviations: BFP, Bolsa Família Program; IPTW, Inverse Probability of Treatment Weighting.

**Table 2.** Mortality rates through receipt of the Bolsa Familia Program, 2011–2015.

| | BFP n = 14,859 | | | Non-BFP n = 14,216 | | |
|---|---|---|---|---|---|---|
| | Events | Person-years at risk | Mortality rate[1] (95% CI) | Events | Person-years at risk | Mortality rate[1] (95% CI) |
| Overall mortality | 403 | 29873.10 | 1349.04 (1223.55, 1487.39) | 587 | 21746.97 | 2699.22 (2489.4,7,2926.66) |
| Natural causes | 238 | 29873.10 | 796.70 (701.65, 904.63) | 468 | 21746.97 | 2152.02 (1965.62, 2356.10) |
| Unnatural causes | 165 | 29873.10 | 552.34 (474.17, 643.38) | 119 | 21746.97 | 547.20 (457.21, 654.90) |

[1]Estimated rates per 100,000.

Abbreviation: BFP, Bolsa Família Program.

**Table 3. Association of Bolsa Família Program participation with mortalities rates, 2011–2015.**

| | Cox model | Competing risk model | |
| --- | --- | --- | --- |
| | Overall mortality<br>N = 761 | Natural causes<br>N = 589 | Unnatural causes<br>N = 223 |
| **Cox adjusted with IPTW[1] (final model)** | HR (95%CI) | HR (95%CI) | HR (95%CI) |
| Non-BFP | 1.00 | 1.00 | 1.00 |
| BFP | 0.82<br>(0.70, 0.95) | 0.34<br>(0.28, 0.41) | 0.90<br>(0.67, 1.19) |
| P value | 0.011 | <0.001 | 0.461 |
| Overall population | 26,093 | | |
| Cox adjusted with Kernel matching[2] | | | |
| Non-BFP | 1.00 | 1.00 | 1.00 |
| BFP | 0.83<br>(0.70, 0.99) | 0.70<br>(0.56, 0.87) | 1.14<br>(0.85, 1.54) |
| P value | 0.042 | 0.001 | 0.373 |
| Overall population | 25,475 | | |

[1]HR estimated with stabilized IPTW given sex, age, race, educational level, household conditions (water supply, waste, sanitation, construction materials), living alone, Brazilian region, location of residence, year of hospitalization or occurrence of interpersonal violence, and year of registration at CadÚnico.

[2]HR estimated with Kernel matching given sex, age, race, educational level, household conditions (water supply, waste, sanitation, construction materials), living alone, Brazilian region, location of residence, year of hospitalization or occurrence of interpersonal violence, and year of registration at CadÚnico.

Abbreviations: BFP, Bolsa Família Program; HR, hazard ratio; CI, confidence interval; IPTW, inverse propensity scores.

the first empirical support that participation in the BFP is associated with a substantial reduction in mortality among individuals exposed to interpersonal violence populations that not only endure the immediate harm of violence but also face long-term risks to their life expectancy. These findings highlight the potential of social protection policies to mitigate the lethal consequences of violence. These findings are particularly striking, given that individuals affected by violence exhibit significantly higher mortality rates compared to the general population, primarily due to violent causes [44].

The BFP might affect mortality in people who have been victims of interpersonal violence through different processes. First, through the conditionalities, these programs are associated with improved access to health services, which in turn may be linked to fewer deleterious changes in the nervous, endocrine, and immune systems related to the stress of violence[2], thereby reducing the risk of mortality. Additionality due to the conditionalities, these individuals may have greater access to other social Brazilian programs which promote inclusion and social protection for people in vulnerability and social risk [45]. Second, the role of the Family Health Strategy in monitoring families who receive benefits through home visits and follow-ups, can be highly relevant in reporting violence, facilitating subsequent treatment, and monitoring other health conditions [46]. Third, the benefit can also promote a greater sense of social inclusion and citizenship by providing access to social, educational, and health services, reducing the vulnerability associated with poverty and violence [5,47]. Fourth, financial support can improve living conditions and nutrition, increase resources, and reduce stress, which can contribute to improving physical and mental health [5,41].

Surprisingly, the BFP was not associated with non-natural causes of death. This finding might have happened because victims of interpersonal violence have an increased risk of dying from violent causes, given that recipients often reside in areas characterized by high rates of urban violence [6], and that a set of systematic public actions, together with the cash transfer, may be required for this already vulnerable population [2]. The findings further indicate that beneficiaries may experience improvements in overall health through the conditionalities of the BFP; however, the program may be insufficient to prevent deaths from violent causes [2].

BFP was also associated with reduced mortality especially among females, possibly by mitigating household-level financial stress [15]. Furthermore, previous research has highlighted the beneficial contribution of CTPs on women's health, particularly in programs such as the BFP, which target pregnant and breastfeeding women [24,48]. Evidence from Low- and Middle-Income Countries (LMICs) indicates that cash transfers improve women's health and increase their use of health services, particularly for maternal and childcare [48].

Another factor that may influence women's health outcomes is the continued presence of the perpetrator within the household. Evidence from previous studies indicates that CTPs can enhance women's autonomy and bargaining power [41], primarily because, in general, women are prioritized in the receipt of the social benefit [23]. Income transfer may alter household power or control dynamics, leading to separation processes when women are exposed to domestic violence [49]. However, it is also possible that perpetrators may appropriate the financial resources provided, thereby sustaining the cycle of domestic violence [41]. This dynamic is reflected in evidence indicating that CTPs have not led to a reduction in femicide rates [49], underscoring the need for complementary public policies specifically aimed at addressing perpetrator behavior and preventing severe forms of violence.

This study has strengths and limitations. To our knowledge, it is the first to examine the association between BFP and overall mortality in victims of interpersonal violence, using nationwide linked datasets. Additionally, since the 100 Million Brazilian Cohort includes data from Brazil's poorest population, this study uniquely highlights the potential contribution of a nationwide social program like BFP in reducing mortality rates among those in socioeconomic vulnerability. In addition, we used a robust analytical approach with a PS-based method, as well as various methodological strategies to check the robustness of our data. Moreover, this study took advantage of estimating survival using a competing risks model. Classical survival analysis models tend to overestimate survival probabilities and underestimate the risks of death, as the presence of competing risks is not considered in the analyses. This highlights the relevance of the competing risks model used in this study. Finally, using administrative data linked with a robust and accurate method reduced recall bias commonly associated with primary data collection.

With regards to limitations, the datasets used in this study were composed of administrative data. Although these data are widely used for many studies [14–18,32,33], they were not designed for research purposes, and some missing data was observed in the covariates. However, the main data used in this study related to outcomes and exposure were fully complete.

Second, considering registration on interpersonal violence from SINAN was not mandatory before 2011, we could only include individuals victimized from 2011 onward, which only enabled us to investigate the short-term contributions of BFP. A longer follow-up may be necessary to fully assess the impact of receiving BFP on mortality long-term, especially among individuals with repeated exposure to violent events. Furthermore, our study only included individuals that seek healthcare services, which likely represent the most severe types of interpersonal violence, potentially resulting in misclassification bias. Additionally, less visible forms of violence, such as psychological violence, may be underreported due to stigma and the difficulty healthcare professionals face in identifying them [50].

Third, we were unable to isolate the contribution of other smaller interventions that might also target low-income families registered at CadÚnico. Similarly, it is difficult to disentangle the role of receiving the benefit itself from the conditionalities attached to it. Therefore, it remains unclear whether the observed associations are attributable to the financial component per se or to behavioral changes required to maintain eligibility for the program. Fourth, although we controlled for sociodemographic covariates, we must consider the influence of unmeasured confounders, especially those related to behavioral factors, such as alcohol abuse. In addition, the covariates were available only at the baseline of the study, which may limit their contribution to the outcome, especially age [39].

Finally, this study may have biases due to the computational complexity involved in the linkage process and the absence of a unique number that identifies individuals across the health and social systems. However, the data used in this study presented high sensitivity and specificity in validation processes which allowed us to find that these errors are probably nondifferential [25].

Considering that we used data from the 100 million Brazilian cohort, which includes individuals with socioeconomic difficulties who applied for social benefits, our findings cannot be generalized for all Brazilian people. The cohort has a specific profile being predominantly composed by women and younger individuals [23]. It is also noteworthy that the profile of the population in this study differed from other studies that used SINAN [51], which is related to the focus of this study on individuals registered in CadÚnico after the violent event.

Our findings revealed the potential contribution that a conditional CTP may have as a public policy for preventing negative health consequences related to interpersonal violence, in a specific population with greater vulnerability due to living in poverty and being victimized by violence. Our study showed that a considerable number of mortalities could be avoided by participation in the BFP. Therefore, governments should integrate cash transfers into efforts to prevent deaths among those vulnerabilized by interpersonal violence, especially conditional programs that improve access to health and social services. Furthermore, based on these findings, it is argued that governments should implement conditional CTPs worldwide, given the significant contribution interpersonal violence has on public health. Conditional CTPs should be advisable for governments to implement than non-conditional programs, as they have the potential to constructive behaviors, mitigating adverse health outcomes associated with interpersonal violence.

## Supporting information

**S1 Appendix. Method A** – Description of the main datasets used in the study. **Method B** – Description of the linkage. **Table A** – Logistic regression to estimate propensity scores for receiving Bolsa Familia Program according to covariates. **Table B** – Association of Bolsa Família Program participation with overall mortality by sex (2011–2015). **Table C** – Association of Bolsa Família Program participation with the outcomes using Poisson model. **Table D** – Association of Bolsa Família Program participation with the outcomes considering overall population who were victim of interpersonal violence (2011–2015). **Table E** – Association of Bolsa Família Program participation with overall mortality considering missing as a category (2011–2015). **Table F** – Crude association of Bolsa Família Program participation with overall mortality (2011–2015). **Table G** – Association of Bolsa Família Program participation with overall mortality using time-varying BFP status (2011–2015). **Table H** – Information Criteria–Based Model Comparison Evaluating the Inclusion of an Interaction Term. **Table I** – Association of Bolsa Família Program participation with overall mortality considering other age categorization in the propensity score estimation (2011–2015). **Fig A** – Distribution of the propensity score in the sample, 2011–2015. (DOCX)

**S1 STROBE Checklist. The STROBE checklist is best used in conjunction with this article (freely available on the Web sites of PLoS Medicine at http://www.plosmedicine.org/, Annals of Internal Medicine at http://www.annals.org/, and Epidemiology at http://www.epidem.com/).** Information on the STROBE Initiative is available at www.strobe-statement.org. (DOC)

## Acknowledgments

We extend our gratitude to the data production team and all collaborators at CIDACS/FIOCRUZ for their efforts in developing the 100 million Brazilian Cohort.

## Author contributions

**Conceptualization:** Flávia Alves, Maurício L Barreto, Vikram Patel, Daiane Borges Machado.

**Data curation:** Camila Bonfim, Flávia Alves.

**Formal analysis:** Camila Bonfim.

**Funding acquisition:** Daiane Borges Machado.

**Investigation:** Daiane Borges Machado.

**Methodology:** Daiane Borges Machado.

**Project administration:** Daiane Borges Machado.

**Resources:** Daiane Borges Machado.

**Supervision:** Maurício L Barreto, Vikram Patel, Daiane Borges Machado.

**Validation:** Camila Bonfim, Flávia Alves, Maurício L Barreto, Vikram Patel, Daiane Borges Machado.

**Visualization:** Camila Bonfim, Flávia Alves, Maurício L Barreto, Vikram Patel, Daiane Borges Machado.

**Writing – original draft:** Camila Bonfim.

**Writing – review & editing:** Flávia Alves, Maurício L Barreto, Vikram Patel, Daiane Borges Machado.

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
