## [Editor Report · Decision Letter 0]

2 Jun 2025

Dear Dr Bonfim,

Thank you for submitting your manuscript entitled "Effect of a large cash transfer on mortality among interpersonal violence victims: a cohort study" for consideration by PLOS Medicine.

Your manuscript has now been evaluated by the PLOS Medicine editorial staff as well as by an academic editor with relevant expertise and I am writing to let you know that we would like to send your submission out for external peer review.

For clinical studies, please upload a copy of your trial study protocol as a supporting information file. The study protocol should be the version submitted for approval to the institutional review board or ethics committee, should include any amendments to the study protocol, as well as the date of their approval by the institutional review or ethics committee. Please also detail any deviations from the study protocol in the Methods section of your manuscript. The editors will consider the protocol and study conduct prior to a final decision for external review.

Please re-submit your manuscript within two working days, i.e. by Jun 04 2025 11:59PM.

Kind regards,

Andreia Cunha, PhD

Senior Editor

PLOS Medicine

---

## [Decision Letter · Decision Letter 1]

10 Sep 2025

Dear Dr Bonfim,

Sincere apologies for the delay in getting back to you with a decision, which was due to challenges in securing the necessary Reviewers. Many thanks for submitting your manuscript "Effect of a large cash transfer on mortality among interpersonal violence victims: a cohort study" (PMEDICINE-D-25-01851R1) to PLOS Medicine. The paper has been reviewed by subject experts and a statistician; their comments are included below and can also be accessed here: [LINK]

As you will see, the reviewers find the work of considerable interest but raise some very important concerns that would need to be fully addressed in a revised manuscript. After discussing the paper with the editorial team and an academic editor with relevant expertise, I'm pleased to invite you to revise the paper in response to the reviewers' comments. However, it seems to us to be essential that the methodological concerns raised by Reviewer 1 are resolved in full. We plan to send the revised paper to some or all of the original reviewers, and we cannot provide any guarantees at this stage regarding publication.

We ask that you submit your revision by Dec 10 2025 11:59PM. However, if this deadline is not feasible, please contact me by email, and we can discuss a suitable alternative.

Don't hesitate to contact me directly with any questions (acunha@plos.org).

Best regards,

Andreia

Andreia Cunha, PhD

Senior editor

PLOS Medicine

acunha@plos.org

Comments from the reviewers:

Reviewer #1: In all, a very interesting manuscript that addresses a relevant public health concern. However, there are some important methodological concerns that preclude us from accepting the manuscript as is. As for some minor comments, we suggest the authors avoid the use of the word "effect" in the title, which may erroneously suggest a causal relationship despite having only these observational data (there was IPTW used, but no causal justifications or assertions were justified in the manuscript itself). As well, results from a Fine-Gray model are presented, but results don't seem to be commented on much (e.g., interpretations of subdistribution hazard ratios, etc.).

One strength of the manuscript is the completeness of the administrative records and their ability to be linked. On line 188 there is a minor typo with a missing "N=". I'm not quite sure how the numbers line up; there seems to be a slight discrepancy (67487 from SINAN + 11286 from SIH - 1174 duplicates results in 77599 as opposed to the 78038 reported). It also would have been helpful to identify the N=14856 of the 29075 individuals in the analysis cohort who received the BFP exposure earlier in the manuscript for reader clarity.

We have two major methodological/statistical comments regarding the manuscript. The first is the degree to which you may ascribe associations as "due to" BFP. Receipt of BFP is based on strict eligibility criteria, nearly all of which are associated with differential outcomes. For instance, the strict eligibility requires /for/ these low-earning individuals are virtually all associated with better outcomes - maintaining attendance, keeping up with vaccinations, and specifying a health and nutrition protocol (implying that they have at least some regular contact with the healthcare system). Thus, a more interesting comparison, though not necessarily possible to evaluate, would be between BFP recipients and those who qualify under BFP eligibility requirements, but who for some reason did /not/ receive the financial benefit. This is to say that it is difficult to believe conclusions as to the benefit of the financial handout itself when receipt of those incentives is tied to behaviors that are likely conducive to lower mortality. Statements such as "The PAF analyses showed that 53% of overall deaths (PAF= 53%, 95% CI 42.1% -330 64.2%) could be avoided by the presence of BFP among those victims of violence," while perhaps accurately applying the definition of PAF mathematically, are thus misleading sociologically/statistically.

The propensity score matching and IPTW procedure was used to try and mitigate these sorts of issues, but their use was not entirely satisfactory. Naturally, one of the largest predictors of mortality is simply age. However, it appears that age was treated as a very coarse categorical variable; a 25 year old and a 59 year old are very different in terms of risk for death due to natural causes. The matching procedure does not appear to be able to account for this. Moreover, we're not even convinced that propensity-based methods are even the ideal in this context. Given that you have essentially complete data from a variety of administrative databases, why not simply treat the participants as one massive cohort for whom you have very good follow-up? You would then be able to use time-varying survival methods to examing adjusted hazard of mortality - the main exposure would be (time-varying) BFP status, adjusting for (time-varying) characteristics such as the ones examined in the present study. Using exact age at study entry would also avoid issues such as the ones raised at the beginning of this paragraph. The use of time-varying methods (for instance, use of BFP exposure as a time-varying covariate) would also avoid potential issues with immortal time bias by having differential classification of patients based on BFP status (i.e., a patient only being eligible to be in the BFP group if they didn't die very shortly after their hospitalization; for these patients, they would serve to inflate the hazard of death among non-BFP patients, simply because these patients didn't live long enough to be "BFP eligible" in the first place).

Regardless, and interesting and well-conducted study; we look forward to reading further revisions and/or responses to these comments.

Reviewer #2: Overall, this paper makes an important contribution to the literature on cash transfers and violence. It was compelling and well done. I have some minor comments below.

- It would be helpful to more detail on which types of violence are reported in SINAN as there are certain types that might be less likely to result in hospitalization. For example, less severe cases like psychological violence or types of violence that people fear having reported like sexual abuse. More discussion is needed about this and how this links to the generalizability of the findings.

- The authors state that the study could only examine the short-term effects of cash transfers on mortality. Because of this, it is likely that this analysis is picking up certain types of mortality and it would be helpful to have some analysis to understand more about that is going on. In particular, looking at modification by age (are these participants older already) and looking at types of mortality including chronic disease outcomes versus other types of morality.

- More discussion is needed about what is meant by "interpersonal violence." I imagine that the effects would be different when we are thinking about violence within a household by a partner or a parent if the cash is going to the household and the perpetrator might receive the money. In these cases, it might be less likely for the victim to benefit from the cash. More discussion is needed about household violence and the implications here as well as how that shows up in the findings.

- Duplicates were excluded but it's important to account for multiple violence experiences as cumulative effects are important with violence experiences. Could you instead make an indicate of multiple experiences and look at the effects among those with 1, 2+ or 0 experiences to see if there are differences there.

- How were the covariates of living conditions and isolation defined?

- More information is needed about the distribution in the length of time receiving a grant after violence exposure and how long it was provided for. Examining modification by these factors would also be important to understand targeting of programs and how soon after violence experiences cash should be provided.

- More information is needed about missingness and why there is no missing data for outcome or exposure. Is this because people with missing data are excluded or not captured here? Were methods like multiple imputation considered?

Minor

- Line 97 typo "the,"

Reviewer #3: This study aims to identify the role of a conditional cash transfer programme in altering trajectories of all-cause, natural, and unnatural mortality among a large sample of Brazilians between 2011 and 2015. Overall, the paper stands to make a contribution to the literature and should be of interest to the journal for publication. However, as written, there are significant limitations that preclude publication in its current form. Generally, the Introduction and Discussion are thin, undercited, and under-explained while the Methods and Results require some additional editing and explanation. Overall, the paper should be edited for English language fluency and be better connected in terms of writing style. Specific comments are below.

Abstract

- Some editing is required to make the Abstract more clear. Specifically, "linked to 100 million Brazilian Cohort baseline from 2011 to 2015" does not make sense as written

Introduction

- Additional explanation of the exact operating definition of interpersonal violence is needed in the Introduction. It remains unclear how broad a definition the study is using. Does it include, for example, gang violence?

- Please further explain "genetic changes associated with the environment" as this is unclear as written.

- There are several examples of editing needed for English language clarity- e.g., "the last one" instead of "the latter"

- Additional details on the BFP are needed for those unfamiliar with the programme. Please provide a paragraph on how it operates as well as the extant literature on its utility in violence and trauma outcomes

- There is no theoretical foundation or theory of change noted- why do authors believe that BFP would alter the trajectory of mortality among victims of interpersonal violence?

Methods

- Please provide additional information on the 100 million Brazilian cohort- though it is cited, more information on these data are required for a critical reading of the paper.

- It should be noted that there is tension between the Introduction (which states that most people do not seek healthcare after violent episodes) and Methods (in which violence is measured via health records in SINAN and SIH. This likely biases the results significantly

- The non-beneficiary group is not well-described in the Methods, which leaves considerable doubt as to the comparability of the beneficiary and non-beneficiary samples. For instance, were the non-beneficiaries labelled as such because they are not eligible for BFP (e.g., their income is too high)? If so, this introduces substantial confounding that is not adequately controlled for using the covariates described in lines 245-247

- "Cox" should be capitalized on line 280

- Like 285 should read "to test whether our results are an artifact…" OR "to test if our results are an artifact"

- There needs to be an explanation of the covariates and how they are measured as well- otherwise readers have no idea what "good household conditions" mean when reading the Results

Results

- Line 304 "in" not "at"

- Line 311 should be "and in Southeast Brazil" .

- Line 312 should be "and were registered"

- Did the authors choose to conduct sex-disaggregated analyses? It is likely that women experiencing intimate partner violence have different trajectories compared to men who experienced community violence due to the differing drivers of each

Discussion

- It is clear that someone different wrote the Discussion than the Methods and Results- please harmonize language, acronyms, and syntax across the paper.

- The authors do not explain the secondary outcome results showing that BFP significantly reduced deaths due to natural, but not unnatural, causes. This, to me, is the more interesting finding since it may suggest that those who take advantage of BFP may improve their health overall through the conditionalities of the programme, but they still likely live in a disproportionately violent area of an already violent country, meaning that BFP in and of itself does not protect from homicide, suicide, or accidental death due to gun, gang, or intimate partner violence. Additional context for these findings would be appreciated.

- A limitation regarding the fact that violence showing up in administrative records is likely the most egregious violence is needed- the study likely has significant misclassification bias that should be noted

- The authors do not engage with the debate around the conditional nature of BFP and whether they recommend CCTs or merely NCCTs- though the proposed mechanisms lend themselves to recommending cash transfer programmes with conditionalities so violence victims are better connected to health and social services.

Reviewer #4: This study has many strengths- a large, national dataset. Use of IPTW and competing risks are strengths of the analyses.

I think the questions is certainly interesting- can CTs reduce mortality AFTER IPV history so not a primary prevention question but a secondary prevention question which is also important but also seems important to know if CTs reduce IPV- among those who get CT (exposure) does outcome (IPV) reduce? which is related but different question.

How did you deal with repeat IPV events over time? Is it correct they were removed per lines 192-193? One concern with removing those with repeated violence is that they are at higher risk of death but those in violent relationships are more likely to experience repeated acts of violence so understanding if the cash reduced future acts is important, although perhaps a separate question- seems they were only 4% of the sample which seems small.

Introduction- seems that describing the difference in sex with IPV is key as women are much more likely to be victims globally. Causes of IPV and intervention effects may differ by sex.

Just to clarify in the results for lines 322-323- :" for unnatural causes of death (HR 0.90, 95% CI: 0.67 - 1.19, p=0.461)" - so no significant decrease in unnatural causes or death only natural? So does this potentially include death resulting from IPV- so if a homicide from an intimate partner would it be included in this death? So per line 361-362 where CCT have been shown to reduce violence 'primarily due to violent causes" it seems in this analysis unnatural causes of death include violent causes but no significant effect is observed here? Is this correct? Why do the authors think this is?

In etable 2 (Poisson) why is the sample size so much smaller than in the cox model eTable 3? Effects are much larger in poisson model which makes me wonder if the samples truly are the same 'population' in the two models even with IPTW?

---

* Please upload any figures associated with your paper as individual TIF or EPS files with 300dpi resolution at resubmission; please read our figure guidelines for more information on our requirements: http://journals.plos.org/plosmedicine/s/figures. While revising your submission, we strongly recommend that you use PLOS's NAAS tool (https://ngplosjournals.pagemajik.ai/artanalysis) to test your figure files. NAAS can convert your figure files to the TIFF file type and meet basic requirements (such as print size, resolution), or provide you with a report on issues that do not meet our requirements and that NAAS cannot fix.

After uploading your figures to PLOS's NAAS tool - https://ngplosjournals.pagemajik.ai/artanalysis, NAAS will process the files provided and display the results in the "Uploaded Files" section of the page as the processing is complete.

If the uploaded figures meet our requirements (or NAAS is able to fix the files to meet our requirements), the figure will be marked as "fixed" above. If NAAS is unable to fix the files, a red "failed" label will appear above.

When NAAS has confirmed that the figure files meet our requirements, please download the file via the download option, and include these NAAS processed figure files when submitting your revised manuscript.

FIGURES AND TABLES

SUPPLEMENTARY MATERIAL

REFERENCES

OBSERVATIONAL STUDIES

* Abstract: Please include the study design, population and setting, number of participants, years during which the study took place (enrollment and follow up), length of follow up, and main outcome measures.

* Please ensure that the study is reported according to the STROBE (or appropriate STOBE extension) guideline (available from: https://www.equator-network.org/reporting-guidelines/strobe) and include the completed STROBE (or STROBE extension) checklist as Supporting Information. Please add the following statement, or similar, to the Methods: "This study is reported as per the Strengthening the Reporting of Observational Studies in Epidemiology (STROBE) guideline (S1 Checklist)." When completing the checklist, please use section and paragraph numbers, rather than page numbers.

* For all observational studies, in the manuscript text, please indicate: (1) the specific hypotheses you intended to test, (2) the analytical methods by which you planned to test them, (3) the analyses you actually performed, and (4) when reported analyses differ from those that were planned, transparent explanations for differences that affect the reliability of the study's results. If a reported analysis was performed based on an interesting but unanticipated pattern in the data, please be clear that the analysis was data driven.

* Please state in the Methods section whether the study had a prospective protocol or analysis plan. If a prospective analysis plan (from your funding proposal, IRB or other ethics committee submission, study protocol, or other planning document written before analyzing the data) was used in designing the study, please include the relevant document(s) with your revised manuscript as a Supporting Information file to be published alongside your study and cite it in the Methods section. A legend for this file should be included at the end of your manuscript. If no such document exists, please make sure that the Methods section transparently describes when analyses were planned, and when/why any data-driven changes to analyses took place. Changes in the analysis, including those made in response to peer review comments, should be identified as such in the Methods section of the paper, with rationale.

MODELLING STUDIES

The following list is derived from Geoffrey P Garnett, Simon Cousens, Timothy B Hallett, Richard Steketee, Neff Walker. Mathematical models in the evaluation of health programmes. (2011) Lancet DOI:10.1016/S0140-6736(10)61505-X:

* If pertinent, please provide a diagram that shows the model structure, including how the natural history of the disease is represented, the process and determinants of disease acquisition, and how the putative intervention could affect the system.

* Please provide a complete list of model parameters, including clear and precise descriptions of the meaning of each parameter, together with the values or ranges for each, with justification or the primary source cited and important caveats about the use of these values noted.

* Please provide a clear statement about how the model was fitted to the data, including goodness-of-fit measure, the numerical algorithm used, which parameter varied, constraints imposed on parameter values, and starting conditions.

* For uncertainty analyses, please state the sources of uncertainties quantified and not quantified [can include parameter, data, and model structure].

* Please provide sensitivity analyses to identify which parameter values are most important in the model. Uncertainty estimates seek to derive a range of credible results on the basis of an exploration of the range of reasonable parameter values. The choice of method should be presented and justified.

* Please discuss the scientific rationale for the choice of model structure and identify points where this choice could influence conclusions drawn. Please also describe the strength of the scientific basis underlying the key model assumptions.

---

## [Decision Letter · Decision Letter 2]

19 Feb 2026

Dear Dr Bonfim,

Sincere apologies for the delay in getting back to you with a decision, which was due to challenges in securing all the necessary advice. Many thanks for submitting your manuscript "Conditional cash transfer and mortality among interpersonal violence victims: a cohort study" (PMEDICINE-D-25-01851R2) to PLOS Medicine. The paper has been reviewed by the original reviewers, including a statistician; their comments are included below and can also be accessed here: [LINK]

As you will see, while two of the Reviewers are now happy with the revision, Reviewer 1 and Reviewer 3 maintain some important concerns that we must ask you to address in full. After discussing the paper with the editorial team and an academic editor with relevant expertise, I'm pleased to invite you to revise the paper in response to the reviewers' comments. Please include in the sensitivity analyses 10-year categories (i.e. 30-39, 40-49, 50-59) and remove any causal language from the Abstract and main text. We plan to send the revised paper to some or all of the original reviewers, and we cannot provide any guarantees at this stage regarding publication.

We ask that you submit your revision by Mar 12 2026 11:59PM. However, if this deadline is not feasible, please contact me by email, and we can discuss a suitable alternative.

Don't hesitate to contact me directly with any questions (acunha@plos.org).

Best regards,

Andreia

Andreia Cunha, PhD

Senior editor

PLOS Medicine

acunha@plos.org

Comments from the reviewers:

Reviewer #1: The authors have provided extensive additional analyses to address previous concerns. While much improved, there still remain statistical issues and inaccuracies. For instance, the revision to add "Therefore, the BFP was associated with a 66% decrease in the rate of natural causes..." regarding the Fine-Gray model is not correct; subdistribution hazards do not correspond to "rates" in that way (note also that epidemiologically, a rate must include a specified time interval in which the rate is calculated).

I am still not satisfied regarding the coarser age stratification wherein a 30 year old and a 59 year old are treated as having the same "age" for purposes of evaluating mortality in the study even though one is essentially double the age of the other (and has multiple times higher incidence rate ratio regarding leading causes of death such as cardiovascular disease). The new age stratification has essentially the same issue (30 year olds and 59 year olds being treated the same) as the previous analysis (25 year olds and 59 year olds being treated the same regarding age).

Finally, although no comparator group is possible in the study, the concern regarding attribution of associations remains. To quote the previous review, "Receipt of BFP is based on strict eligibility criteria, nearly all of which are associated with differential outcomes. For instance ... maintaining attendance, keeping up with vaccinations, and specifying a health and nutrition protocol (implying that they have at least some regular contact with the healthcare system). ... This is to say that it is difficult to believe conclusions as to the benefit of the financial handout itself when receipt of those incentives is tied to behaviors that are likely conducive to lower mortality." More explicitly discussing this difficulty would make for a less misleading discussion.

Reviewer #2: The authors have sufficiently addressed my comments.

Reviewer #3: THe authors have adequately addressed my concerns. One note- the authors say on line 464 that "BFP was also associated with reduced mortality especially among females, a group at higher risk for interpersonal violence". This is not true- interpersonal violence writ large (which encompasses all forms of violent victimization- homicide, assault, etc.) affects far more men than women. However, women are much more likely to experience violence from an intimate partner, to experience sexual violence, and to be murdered by a romantic partner. The citation used here (Sardinha et al) does not address violent victimization among men and so is an inappropriate citation for the claim made. I suggest qualifying this statement by saying something about how BFP reduces mortality among females, likely by reducing financial strain within a family unit.

Reviewer #4: Overall I think the authors have ben very responsive to the comments of reviewers and this is an important analysis to publish. No further comments.

---

* Please upload any figures associated with your paper as individual TIF or EPS files with 300dpi resolution at resubmission; please read our figure guidelines for more information on our requirements: http://journals.plos.org/plosmedicine/s/figures. While revising your submission, we strongly recommend that you use PLOS's NAAS tool (https://ngplosjournals.pagemajik.ai/artanalysis) to test your figure files. NAAS can convert your figure files to the TIFF file type and meet basic requirements (such as print size, resolution), or provide you with a report on issues that do not meet our requirements and that NAAS cannot fix.

After uploading your figures to PLOS's NAAS tool - https://ngplosjournals.pagemajik.ai/artanalysis, NAAS will process the files provided and display the results in the "Uploaded Files" section of the page as the processing is complete.

If the uploaded figures meet our requirements (or NAAS is able to fix the files to meet our requirements), the figure will be marked as "fixed" above. If NAAS is unable to fix the files, a red "failed" label will appear above.

When NAAS has confirmed that the figure files meet our requirements, please download the file via the download option, and include these NAAS processed figure files when submitting your revised manuscript.

FIGURES AND TABLES

SUPPLEMENTARY MATERIAL

REFERENCES

OBSERVATIONAL STUDIES

* Abstract: Please include the study design, population and setting, number of participants, years during which the study took place (enrollment and follow up), length of follow up, and main outcome measures.

* Please ensure that the study is reported according to the STROBE (or appropriate STOBE extension) guideline (available from: https://www.equator-network.org/reporting-guidelines/strobe) and include the completed STROBE (or STROBE extension) checklist as Supporting Information. Please add the following statement, or similar, to the Methods: "This study is reported as per the Strengthening the Reporting of Observational Studies in Epidemiology (STROBE) guideline (S1 Checklist)." When completing the checklist, please use section and paragraph numbers, rather than page numbers.

* For all observational studies, in the manuscript text, please indicate: (1) the specific hypotheses you intended to test, (2) the analytical methods by which you planned to test them, (3) the analyses you actually performed, and (4) when reported analyses differ from those that were planned, transparent explanations for differences that affect the reliability of the study's results. If a reported analysis was performed based on an interesting but unanticipated pattern in the data, please be clear that the analysis was data driven.

* Please state in the Methods section whether the study had a prospective protocol or analysis plan. If a prospective analysis plan (from your funding proposal, IRB or other ethics committee submission, study protocol, or other planning document written before analyzing the data) was used in designing the study, please include the relevant document(s) with your revised manuscript as a Supporting Information file to be published alongside your study and cite it in the Methods section. A legend for this file should be included at the end of your manuscript. If no such document exists, please make sure that the Methods section transparently describes when analyses were planned, and when/why any data-driven changes to analyses took place. Changes in the analysis, including those made in response to peer review comments, should be identified as such in the Methods section of the paper, with rationale.

MODELLING STUDIES

The following list is derived from Geoffrey P Garnett, Simon Cousens, Timothy B Hallett, Richard Steketee, Neff Walker. Mathematical models in the evaluation of health programmes. (2011) Lancet DOI:10.1016/S0140-6736(10)61505-X:

* If pertinent, please provide a diagram that shows the model structure, including how the natural history of the disease is represented, the process and determinants of disease acquisition, and how the putative intervention could affect the system.

* Please provide a complete list of model parameters, including clear and precise descriptions of the meaning of each parameter, together with the values or ranges for each, with justification or the primary source cited and important caveats about the use of these values noted.

* Please provide a clear statement about how the model was fitted to the data, including goodness-of-fit measure, the numerical algorithm used, which parameter varied, constraints imposed on parameter values, and starting conditions.

* For uncertainty analyses, please state the sources of uncertainties quantified and not quantified [can include parameter, data, and model structure].

* Please provide sensitivity analyses to identify which parameter values are most important in the model. Uncertainty estimates seek to derive a range of credible results on the basis of an exploration of the range of reasonable parameter values. The choice of method should be presented and justified.

* Please discuss the scientific rationale for the choice of model structure and identify points where this choice could influence conclusions drawn. Please also describe the strength of the scientific basis underlying the key model assumptions.

---

## [Decision Letter · Decision Letter 3]

31 Mar 2026

Dear Dr. Bonfim,

I am writing on behalf of my colleague Dr. Andreia Cunha. Thank you very much for re-submitting your manuscript "Conditional cash transfer and mortality among interpersonal violence victims: a cohort study" (PMEDICINE-D-25-01851R3) for review by PLOS Medicine.

I have discussed the paper with my colleagues and the academic editor and it was also seen again by one reviewer. I am pleased to say that provided the remaining editorial and production issues are dealt with we are planning to accept the paper for publication in the journal.

[LINK]

We look forward to receiving the revised manuscript by Apr 07 2026 11:59PM.

Sincerely,

Alison Farrell, PhD

Senior Editor

PLOS Medicine

plosmedicine.org

Requests from Editors:

* Please ensure that the statements in the manuscript metadata match those in the manuscript, specifically the Data Availability statement.

* Please note that we ask authors to describe all restrictions to data access, therefore please remove the 'reasonable request' statement in the DAS and describe the process for obtaining access to data.

* Please remove the funders from the Acknowledgements statement.

* Statistical reporting: Please revise throughout the manuscript, including tables and figures.

- Please report statistical information as follows to improve clarity for the reader ""22% (95% CI [13,28]; p</=)"".

- Please separate upper and lower bounds with commas instead of hyphens as the latter can be confused with reporting of negative values.

- Please repeat statistical definitions (HR, CI etc.) for each set of parentheses.

* In the abstract, please include the important dependent variables that are adjusted for in the analyses.

* In the Abstract (line 45, missing 'the') and Author Summary (last sentence), please review the grammar.

* In the Abstract, please remove the causal statement(s) and use associative language (e.g. line 52 should be revised).

* Associative language should be used throughout.

* Please ensure that all abbreviations are defined at first use throughout the text.

* Please confirm that all numbers presented in the abstract are present and identical to numbers presented in the main manuscript text.

* The funding statement should include: specific grant numbers, initials of authors who received each award, URLs to sponsors’ websites. Also, please state whether any sponsors or funders (other than the named authors) played any role in study design, data collection and analysis, the decision to publish, or preparation of the manuscript. If they had no role in the research, include this sentence: “The funders had no role in study design, data collection and analysis, decision to publish, or preparation of the manuscript.”

The URLs are missing from the funding statement.

* Please update the STROBE checklist if necessary and upload it as a supporting information file (not as 'other').

* PLOS defines the “minimal data set” to consist of the data set used to reach the conclusions drawn in the manuscript with related metadata and methods, and any additional data required to replicate the reported study findings in their entirety. Authors do not need to submit their entire data set, or the raw data collected during an investigation. Please submit the following data:

The values behind the means, standard deviations and other measures reported;

The values used to build graphs;

The points extracted from images for analysis.

"* The Data Availability Statement (DAS) requires revision. For each data source used in your study:

* Please add the following statement, or similar, to the Methods: ""This study is reported as per the Strengthening the Reporting of Observational Studies in Epidemiology (STROBE) guideline (S1 Checklist).""

"* Did your study have a prospective protocol or analysis plan? Please state this (either way) early in the Methods section.

* For all observational studies, in the manuscript text, please indicate: (1) the specific hypotheses you intended to test, (2) the analytical methods by which you planned to test them, (3) the analyses you actually performed, and (4) when reported analyses differ from those that were planned, transparent explanations for differences that affect the reliability of the study's results. If a reported analysis was performed based on an interesting but unanticipated pattern in the data, please be clear that the analysis was data-driven.

Comments from Reviewers:

Reviewer #1: We are satisfied with the additional analyses (now in supplement 9) and revisions regarding statistical language.

[LINK]

---

## [Editor Report · Decision Letter 4]

16 Apr 2026

Dear Dr Bonfim,

On behalf of my colleagues and the Academic Editor, Dr Mark Tomlinson, I am pleased to inform you that we have agreed to publish your manuscript "Conditional cash transfer and mortality among interpersonal violence victims: a cohort study" (PMEDICINE-D-25-01851R4) in PLOS Medicine.

PRESS

Sincerely,

Andreia Cunha, PhD

Senior Editor

PLOS Medicine